# Okadaic Acid Activates JAK/STAT Signaling to Affect Xenobiotic Metabolism in HepaRG Cells

**DOI:** 10.3390/cells12050770

**Published:** 2023-02-28

**Authors:** Leonie T. D. Wuerger, Felicia Kudiabor, Jimmy Alarcan, Markus Templin, Oliver Poetz, Holger Sieg, Albert Braeuning

**Affiliations:** 1Department of Food Safety, German Federal Institute for Risk Assessment, Max-Dohrn-Str. 8-10, 10589 Berlin, Germany; 2NMI Natural and Medical Sciences Institute, Markwiesenstraße 55, 72770 Reutlingen, Germany; 3SIGNATOPE GmbH, Markwiesenstraße 55, 72770 Reutlingen, Germany

**Keywords:** okadaic acid, HepaRG cells, CYP enzymes, NF-κB, JAK/STAT, inflammation

## Abstract

Okadaic acid (OA) is a marine biotoxin that is produced by algae and accumulates in filter-feeding shellfish, through which it enters the human food chain, leading to diarrheic shellfish poisoning (DSP) after ingestion. Furthermore, additional effects of OA have been observed, such as cytotoxicity. Additionally, a strong downregulation of the expression of xenobiotic-metabolizing enzymes in the liver can be observed. The underlying mechanisms of this, however, remain to be examined. In this study, we investigated a possible underlying mechanism of the downregulation of cytochrome P450 (CYP) enzymes and the nuclear receptors pregnane X receptor (PXR) and retinoid-X-receptor alpha (RXRα) by OA through NF-κB and subsequent JAK/STAT activation in human HepaRG hepatocarcinoma cells. Our data suggest an activation of NF-κB signaling and subsequent expression and release of interleukins, which then activate JAK-dependent signaling and thus STAT3. Moreover, using the NF-κB inhibitors JSH-23 and Methysticin and the JAK inhibitors Decernotinib and Tofacitinib, we were also able to demonstrate a connection between OA-induced NF-κB and JAK signaling and the downregulation of CYP enzymes. Overall, we provide clear evidence that the effect of OA on the expression of CYP enzymes in HepaRG cells is regulated through NF-κB and subsequent JAK signaling.

## 1. Introduction

Okadaic acid (OA) is a lipophilic marine biotoxin produced by dinoflagellates. It accumulates in the fatty tissue of filter-feeding shellfish and can lead to diarrheic shellfish poisoning (DSP) after ingestion of contaminated shellfish [1]. With the rise in the occurrence of so-called harmful algae blooms due to climate change and industrial waste, the abundance of DSP-toxin-producing dinoflagellates is also rising. DSP is especially a problem in Europe and Japan, but also occurs in other countries around the world [2]. Leading symptoms include diarrhea, stomach pain, and vomiting. To prevent DSP, the European Union implemented a limit of 160 OA equivalents/kg shellfish meat, based on acute toxic effects, such as diarrhea [3]. Moreover, there are multiple reported properties of OA, such as cytotoxicity [4,5,6] and embryotoxicity in vitro [7,8]. However, no embryotoxic effect of OA in humans has been evaluated so far. It was shown to pass the placental barrier in mice [9]. The toxin also acts as a tumor promotor in various organs [10,11] and a correlation between frequent consumption of contaminated shellfish and colon cancer was reported [12,13,14].

OA was first discovered in the black sponge *Halichondria okadaic,* and in 1981 the structure was published [15]. OA was very early reported as a phosphatase inhibitor; its main target on the molecular level is protein phosphatase 1 and 2A [16], but it is also able to target other serine/threonine phosphatases. Therefore, OA exposure can lead to hyperphosphorylation, which can lead to modifications in signal transduction [17].

Because of DSP symptoms, most previous studies focused on the intestine; however, liver toxicity was also reported in mice [18]. Dietrich et al. were able to demonstrate a disruption of the tight junction proteins, such as claudins and occludin, in the intestine after OA exposure at food-relevant concentrations [19]. OA is therefore able to pass the mechanical barrier of the intestine. Furthermore, we recently demonstrated an effect of OA on xenobiotic metabolism in liver cells [20]. In a recent publication, we were able to show that OA is able to downregulate several cytochrome P450 (CYP) enzymes at the RNA and protein level and to decrease their activity. Furthermore, OA is able to influence the expression of several transporter proteins and transcription factors associated with xenobiotic metabolism, which indicates that OA is able to interact with the biochemical barrier in the liver. However, the underlying mechanism of CYP regulation remains to be elucidated.

It has already been demonstrated that OA is able to activate nuclear factor kappa B (NF-κB) in different cell types [21,22,23,24,25]. NF-κB activation leads to elevated levels of pro-inflammatory cytokines [26], and therefore plays a major role in inflammation (cp. Figure 1). Keller et al. demonstrated that elevated cytokine levels in primary human hepatocytes activate the Janus kinase/signal transducers and activators of transcription (JAK/STAT) signaling pathway to affect the expression of xenobiotic-metabolizing enzymes [27].

The JAK/STAT pathway is involved in several inflammatory and autoimmune diseases, but also in cell proliferation, differentiation, and apoptosis. It is therefore an essential pathway in human cells [28].

In this study, we provide an advanced characterization of the OA-mediated inflammatory response. We propose an activation pathway involving NF-κB and JAK/STAT signaling (cp. Figure 1) and determine if OA is able to activate the JAK/STAT signaling pathway in HepaRG cells through activation of NF-κB and if activation of the pathway is connected to CYP downregulation.

We used differentiated HepaRG cells and incubated them with OA and several inhibitors, targeting several steps in the proposed activation pathway. HepaRG cells originate from human hepatocarcinoma. With the addition of DMSO to the cell culture medium, they can be differentiated into hepatocyte- and biliary-epithelium-like cells. Differentiated HepaRG cells express a variety of liver-specific enzymes and also xenobiotic metabolizing enzymes at a similar level to primary hepatocytes. Therefore, they are a very common model in studies examining xenobiotic metabolism in vitro [29,30].

## 2. Materials and Methods

### 2.1. Chemicals

OA was purchased from Enzo Life Sciences GmbH (Loerrach, Germany). JSH-23 and Methysticin were purchased from Merck KGaA (Darmstadt, Germany) and Decernotinib and Tofacitinib were obtained from Selleck Chemicals (Munich, Germany). All other standard chemicals and materials were purchased from Sigma-Aldrich (Taufkirchen, Germany) or Roth (Karlsruhe, Germany) in the highest available purity.

### 2.2. Cell Cultivation

HepaRG cells (Biopredic International, Saint-Grégoire, France) were cultivated at 37 °C for 14 days in William’s E medium supplemented with 10% fetal bovine serum (FBS), 5 μg/mL insulin (medium and both supplements from PAN-Biotech GmbH, Aidenbach, Germany), 50 μM hydrocortisone hemisuccinate (Sigma-Aldrich, Taufkirchen, Germany), 100 U/mL penicillin, and 100 μg/mL streptomycin (Capricorn Scientific, Ebsdorfergrund, Germany), according to established protocols. After 14 days, the medium was further supplemented with 1% dimethyl sulfoxide (DMSO) for 2 days, and afterwards, the DMSO content was increased to 1.7% for another 12 days. Then, the medium was changed to serum free assay medium (SFM), which consisted of William’s E medium without phenol red (PAN-Biotech GmbH, Aidenbach, Germany), containing 100 U/mL penicillin and 100 μg/mL streptomycin, 2.5 μM hydrocortisone hemisuccinate, 10 ng/mL human hepatocyte growth factor (Biomol GmbH, Hamburg, Germany), 2 ng/mL mouse epidermal growth factor (Sigma-Aldrich, Taufkirchen, Germany) and 0.5% DMSO. This treatment medium was adapted from Klein et al. [31].

Cultivation of HepG2 cells (ECACC, Porton Down, UK) was performed as described before [32]. HepG2 cells were used for development of the confocal microscopy method.

The human embryonic kidney cell line HEK-T was obtained from the European Collection of Cell Cultures (ECACC, Porton Down, UK). The cells were cultured in high-glucose Dulbecco’s modified Eagle’s medium (DMEM, Pan-Biotech GmbH, Aidenbach, Germany) supplemented with 10% fetal calf serum (Pan-Biotech GmbH, Aidenbach, Germany), 100 U/mL penicillin, and 100 g/mL streptomycin (PAA Laboratories GmbH, Pasching, Austria). The cells were passaged every 2–4 days (80–90% confluence) and seeded at 5 × 10^4^ cells/cm^2^ in 96-well plates.

### 2.3. Gene Expression Analysis

In total, 2 × 10^5^ cells/well were seeded in 6-well plates and treated with 11, 33, and 100 nM OA. Furthermore, they were treated with 30 µM JSH-23 and 30 µM Methysticin, two NF-κB activation inhibitors, and 40 µM Decernotinib and 40 µM Tofacitinib, two JAK inhibitors, for 24 h each. Afterwards, the gene expression of several cytokines, SOCS3, and 3 representative CYP enzymes was determined using quantitative real-time reverse transcriptase polymerase chain reaction (qPCR), which was performed as described before [20]. The used primers can be found in Appendix A.

### 2.4. Western Blotting

The STAT1 and STAT3 proteins were analyzed using Western blotting. Cell cultivation in 6-well plates and the incubation was performed as described above for the gene expression analysis. After 24 h incubation, the cells were harvested in PBS after a washing step. The cells were centrifuged (5 min, 2000× *g*, 4 °C) and the supernatant was discarded. The proteins were then isolated from the pellets using RIPA lysis buffer (pH 7.5; 50 mM Tris-HCl, 150 mM NaCl, 2 µM EGTA, 0.1% sodium dodecyl sulfate (SDS), and 0.5% desoxycholic acid) containing 1:50 protease inhibitor (Complete Protease Inhibitor Cocktail Tablets, Roche, Mannheim, Germany) and 1% Triton X-100. The cell pellets were then rotated at 4 °C for 15 min and homogenized using an ultrasonic homogenizer (Sonopuls HD 2070, BANDELIN electronic GmbH & Co. KG, Berlin, Germany, 25% power, pulse 2). The homogenized lysates were then centrifuged at 4 °C and 13,200× *g* for 30 min. The cell pellet was discarded. The protein content in the supernatants was determined using Bradford assay according to the manual of the Biorad protein assay (Bio-Rad Laboratories GmbH, Feldkirchen, Germany) against a bovine serum albumin standard curve. The samples were diluted with MilliQ water, mixed with 5× Laemmli buffer (1:5; 320 mM Tris-HCl, 15% SDS (*w*/*v*), 35% glycerine, 0.5% bomophenol blue, and 25% 2-mercaptoethanol) and denatured at 96 °C for 5 min. SDS polyacrylamide gel electrophoresis was performed using a 5% stacking gel and a 10% separation gel, followed by a transfer of the separated proteins to a nitrocellulose membrane (Amersham™ Protran™ Premium NC, Cytiva Europe GmbH, Freiburg, Germany) using the wet blot method. Unspecific binding sites were then blocked using 5% milk powder in Tris-buffered saline with Tween-20 (TBS-T, pH 7.6, 20 mM Tris-HCl, 137 mM NaCl, and 0.1% Tween-20) for 1 h at room temperature. The first antibody against phosphoSTAT1 (Phospho-STAT1 (Tyr701) Recombinant Rabbit Monoclonal Antibody (15H13L67), Thermo Fisher Scientific, Waltham, MA, USA, dilution factor 1:1000) or phosphoSTAT3 (Phospho-STAT3 (Ser727) Recombinant Rabbit Monoclonal Antibody (SY24-09), Thermo Fisher Scientific, Waltham, MA, USA, dilution factor 1:1000) was then incubated overnight at 4 °C. The membrane was then washed 3× with TBS-T and incubated with the secondary antibody (Anti-rabbit, HRP-conjugated, R&D systems, Minneapolis, MN, USA, dilution factor: 1:1000) for 1 h at room temperature. For detection, the membrane was incubated with the SuperSignal West Fento Maximum Sensitivity Substrate Kit according to the included protocol (Thermo Fisher Scientific, Waltham, MA, USA). The chemiluminescence was detected using a Molecular Imager Versadoc MP 4000 (BioRad Laboratories GmbH, Feldkirchen, Germany). As loading control, GAPDH was afterwards stained as well. Therefore, the membrane was incubated with the Anti-GAPDH antibody [6C5] (abcam, Cambridge, UK, dilution factor 1:7500) for 1 h at room temperature. For detection, Sheep anti-Mouse-IgM-HRP antibody was incubated for 1 h (HRP-conjugated, Seramun Diagnostica GmbH, Heidesee, Germany, dilution factor 1:10,000), followed by protein detection as described above. For quantification of the band intensities, ImageLab 6.0.1 software was used. Intensities were then referred to the loading control GAPDH and normalized to the respective solvent control.

### 2.5. Confocal Microscopy

First, 0.1 × 10^6^ cells were seeded in 12-well plates containing a coverslip in each well. After differentiation, the cells were treated with 33 nM and 100 nM OA, 40 µM Decernotinib and 40 µM Tofacitinib, and the respective solvent control for 24 h. Cells were fixed for 20 min using 3.7% formaldehyde and permeabilized for 10 min with 0.5% Triton-X in PBS. Afterwards, they were incubated with DAPI (AppliChem, Darmstadt, Germany), which was diluted 1:1000 in PBS for 20 min at room temperature. Actin was stained with the ActinGreen™ 488 ReadyProbes™ reagent (Invitrogen™, Waltham, MA, USA). Two drops of ActinGreen were dissolved in 1 mL PBS and incubated on the cells at room temperature for 30 min. Afterwards, the samples were blocked with blocking solution (1% BSA in PBS-T) for 1 h at room temperature. The primary antibody, Anti-NF-kB p65 antibody ab16502 (Abcam, Cambridge, UK) for NF-κB and Phospho-STAT3 (Ser727) Recombinant Rabbit Monoclonal Antibody (SY24-09) (Invitrogen™, Waltham, MA, USA) for phosphoSTAT3 was then diluted 1:1000 in blocking solution and incubated on the cells at 4 °C overnight. Afterwards, the secondary antibody, Goat anti-Rabbit IgG (H+L) Cross-Adsorbed Secondary Antibody, Alexa Fluor™ 633 (Life Technologies™, Carlsbad, CA, USA) was incubated for 1 h at room temperature. The coverslips with the cells were then flipped onto a glass slide containing a drop of Vectashield^®^ HardSet™ Antifade Mounting Medium (Vector Laboratories, Newark, New Jersey, CA, USA) and dried overnight at 4 °C.

Fluorescence was detected using the confocal laser scanning microscope LSM 700 (Zeiss, Oberkochen, Germany) at ex wavelengths of 405 nm (DAPI, blue), 488 nm (ActinGreen, green), and 633 nm (NF-κB or phosphoSTAT3, red). Z-stacks spanning through the entire cell layer were recorded at 63× magnification.

For analysis, 6 images of each condition were collected and analyzed using the ImageJ 1.53e software (Laboratory for Optical and Computational Instrumentation (LOCI)) at the University of Wisconsin–Madison, Madison, WI, USA). For quantification of the colocalization, the nuclear fluorescence ratio of the target protein to the DAPI signal was determined for each single nucleus. The nuclei were marked as region of interest (ROI) and the fluorescence intensity of the red and blue channels in each ROI was determined separately. A ratio of red/blue signal intensity was then formed and normalized to the mean of the solvent controls. The nuclei were then divided into groups based on their signal intensity.

### 2.6. Protein Quantification

Proteins were quantified using the Luminex xMAP^®^ technology. In total, 2 × 10^5^ cells/well were seeded in 6-well plates and treated with 11, 33, and 100 nM OA. Furthermore, they were treated with 30 µM JSH-23, 30 µM Methysticin, and 40 µM Decernotinib or 40 µM Tofacitinib for 24 h. After incubation, the cell culture supernatant was collected and frozen at −80 °C until further use. After thawing, the supernatants were treated as described in the instruction manual of the ProcartaPlex™ Basiskit, human (Invitrogen™, Waltham, MA, USA) using the IL-1 alpha Human ProcartaPlex™ Simplex Kit, IL-1 beta Human ProcartaPlex™ Simplex Kit, IL-6 Human ProcartaPlex™ Simplex Kit, IL-8 (CXCL8) Human ProcartaPlex™ Simplex Kit, IL-12 p70 Human ProcartaPlex™ Simplex Kit, and the TNF alpha Human ProcartaPlex™ Simplex Kit (all Invitrogen™, Waltham, MA, USA). The plates were measured on a FLEXMAP 3D^®^ instrument (Luminex, Austin, TX, USA). The data were evaluated in Origin 2017 (OriginLab, Northampton, MA, USA) as described in the instruction manual.

DigiWest^®^ was performed as described before [33]. DigiWest^®^ was performed for p100/p52 and RelB subunits of NF-κB, IκB kinase (IκBα), IκB kinase subunit IKKβ, Jak1, and Jak2. Three independent replicates were combined before measuring.

### 2.7. PXR and RXRα Transactivation Assay

Transactivation assays were conducted as previously described [34]. Briefly, 24 h after seeding, HEK-T cells were transiently transfected using TransIT-LT1 (Mirus Bio, Madison, WI, USA) according to the manufacturer’s protocol. For each well, the transfection mixture contained 40 ng pGAL4-(UAS)5-TK-luc, 40 ng pGAL4-hPXR-LBD, 1 ng pcDNA3-Rluc for PXR assay and 40 ng pGAL4-(UAS)5-TK-luc, 40 ng pCMX-GAL4-hRXRα, and 1 ng pcDNA3-Rluc for RXRα assay. pcDNA3-Rluc was used as an internal control for normalization. Four to six hours after transfection, the cells were incubated with different concentrations of OA dissolved in culture medium (0.1% MeOH). PXR agonist SR12813 (10 µM) and RXRα agonist CD2608 (100 nM) were used as positive controls. After 24 h, the culture medium was removed and the cells were lysed after addition of 50 µL lysis buffer (100 mM potassium phosphate with 0.2% (*v*/*v*) Triton X-100, pH 7.8) for 15 min on an orbital shaker. After centrifugation (5 min, 2000× *g*), 5 µL of the supernatant was analyzed for luciferase activity as previously described [35]. Firefly luciferase values were normalized to Renilla luciferase values and expressed as fold-induction normalized against solvent control.

### 2.8. Statistical Analysis

Statistical analysis was performed using the software SigmaPlot (SYSTAT Software Inc., San Jose, CA, USA). To determine the statistical significance, one-way ANOVA followed by Dunnett’s post-hoc test (* *p* < 0.05; ** *p* < 0.01; *** *p* < 0.001) was performed to compare the different sample groups. For confocal microscopy, Wilcoxon rank-sum test (* *p* < 0.05; ** *p* < 0.01; *** *p* < 0.001) was performed to compare the sample groups against a control.

## 3. Results

In this study, we chose three different OA concentrations based on previously published results of cell viability testing [20], where we incubated HepaRG cells with OA concentrations between 5 and 500 nM for 24 h. We chose 11 nM, 33 nM, and 100 nM as non-toxic concentrations. In this study, we also added different inhibitors, targeting different points of the proposed signaling pathway. JSH-23 and Methysticin inhibit the activation of NF-κB, while Decernotinib and Tofacitinib inhibit the activation of JAK.

Figure 2A shows the analysis of the RNA expression of some CYP enzymes. *CYP1A1, CYP2B6,* and *CYP3A4* were selected as representative CYP enzymes and analyzed using qPCR. As previously shown in Wuerger et al. [20], OA was able to strongly inhibit the expression of these CYPs. Furthermore, we investigated the effect of OA on the transcription factors PXR and RXRα. Those factors directly influence the expression of various CYPs, importantly *CYP3A4* and *CYP2B6*. As shown in Figure 2B, OA was also able to strongly downregulate the RNA expression of the two transcription factors PXR and RXRα in HepaRG cells, as well as their transcriptional activity, as shown in HEK-T cells using reporter gene transactivation assays. In summary, OA was able to strongly downregulate the expression of *CYP1A1, CYP2B6,* and *CYP3A4* and also able to downregulate the expression and activity of the transcription factors PXR and RXRα.

Based on these data and information from the literature as detailed in the introduction section, we hypothesized a possible mechanism induced by OA involving the activation of NF-κB- and subsequent JAK/STAT signaling pathways (Figure 1). Activation of NF-κB consequently activates the expression of several cytokines released into the surrounding cell culture medium. Activated cytokine receptors recruit JAKs, which become activated to subsequently activate STAT proteins, which translocate into the nucleus, where they can act as transcription factors for SOCS3 [36,37], or for other genes related to xenobiotic metabolism. SOCS3 is a direct target of activated STAT3 and able to inhibit JAK/STAT, especially if activated via Interleukin-6- (IL-6) dependent signaling in a negative feedback loop [38].

To verify the activation of NF-κB by OA, we developed a confocal microscopy method. After activation of NF-κB, p65 translocates into the nucleus, which can be visualized using a specific primary antibody against p65 (Figure 3A). The activation and translocation of NF-κB by OA was observed in HepG2 cells (Appendix A). Figure 3A shows the activation of NF-κB by OA in a concentration-dependent manner in HepaRG cells. To visualize a colocalization of p65 and the nucleus, a cross-section through the entire cell layer was visualized. Figure 3B shows an increase in activated NF-κB with an increase of the OA concentration. Furthermore, DigiWest analysis, an advanced digital fluorescence-based Western blotting method, showed that the protein expression of another NF-κB subunit, RelB, was strongly upregulated after OA incubation. Furthermore, protein expression of IκBα and IKKβ were downregulated by OA, which is shown in Figure 3C. In summary, Figure 3 shows the activation of NF-κB in HepaRG cells after incubation with OA. The NF-κB-inhibitors JSH-23 and Methysticin were also evaluated in combination with OA but showed no effect on the translocation (Appendix A).

Activation of NF-κB leads to expression and release of cytokines. We were able to show that the mRNA expression of several interleukins was upregulated in HepaRG cells upon exposure to OA. To show a direct involvement of NF-κB activation through OA with these effects, we then added NF-κB inhibitors. The NF-κB inhibitors were able to reverse the observed effect of OA on the ILs (Figure 4A). This suggests an involvement of NF-κB signaling in IL mRNA expression. To verify the effects of OA on the RNA expression of ILs in HepaRG cells, we then investigated the protein expression of IL-6 and IL-8 using Luminex xMAP^®^ technology. Figure 4B shows the release of IL-6 and IL-8 into the cell culture supernatant. IL-1α, IL-1β, IL-12, and TNF-α were also analyzed. Results for IL-1α can be found in the Appendix A, while the levels of the other cytokines did not exceed the LLOQ (IL-1β: 2.76 pg/mL; IL-12: 7.74 pg/mL; TNFα: 6.15 pg/mL). OA exposure clearly led to a concentration-dependent release of cytokines from HepaRG cells. Furthermore, combined incubation with OA and the NF-κB inhibitor Methysticin was able to counteract the cytokine release induced by OA, which is shown in Figure 4C. Results of cytokine release after incubation with JAK inhibitors can also be found in the Appendix A. The addition of NF-κB inhibitors further influenced the RNA expression of the CYP enzymes *CYP1A1, CYP2B6,* and *CYP3A4*. Figure 4D shows that their expression is decreased with OA, which could be partially reversed with the NF-κB inhibitors, thus indicating that OA exerts its effects on the CYPs via NF-κB activation.

Figure 5 shows the activation of the JAK/STAT signaling pathway in HepaRG cells after exposure to OA. Figure 5A was obtained using a similar method as in Figure 3, with a specific antibody against phosphorylated STAT3 (phosphoSTAT3), the activated form of STAT3. Figure 5A,B show that more activated STAT3 is present in the cells incubated with OA than in the solvent control. Furthermore, the JAK inhibitors Decernotinib and Tofacitinib (images shown in Appendix A) were inhibiting STAT3 activation by OA (Figure 5F). This result was further validated using Western blotting. Figure 5C shows a representative Western blot using the antibody against phosphoStat3 (pSTAT3). A strong activation of STAT3 was observed following OA incubation in HepaRG cells. We were able to reverse this effect using the JAK inhibitor Tofacitinib. On the contrary, no activation was detected for STAT1 using Western blotting (Appendix A). Furthermore, using DigiWest, we could show a higher protein expression of JAK1 upon OA incubation in HepaRG cells, whereas JAK2 expression was downregulated (Figure 5E). These results point towards an activation of JAK1 and subsequent activation of STAT3 by OA. Additionally, we again evaluated the RNA expression of CYP enzymes *CYP1A1, CYP2B6,* and CYP3A4 after OA exposure in combination with the JAK inhibitors. The strong CYP downregulation mediated by OA was significantly reversed using the inhibitors (Figure 5D), thus demonstrating that JAK signaling is involved in mediating the effects of OA.

## 4. Discussion

We recently demonstrated an effect of OA on the metabolic barrier in the liver: OA is able to downregulate the expression of several xenobiotic-metabolizing enzymes on the RNA and protein levels as well as their enzymatic activity (this work, [20]). Many studies have shown that inflammation can alter the metabolic function of xenobiotic metabolism [27,39]. While most studies have focused on only one inflammatory effector (eg., NF-κB activation or Cytokine release), our study provides deeper insight into the molecular mechanism of OA-mediated inflammatory response.

Firstly, we evaluated the expression and transcriptional activity of PXR and RXRα, two transcription factors regulating CYP expression. Both endpoints were significantly downregulated upon exposure of cells to OA. We postulated a possible mechanism as underlying signaling pathway for the altered expression of xenobiotic-metabolizing enzymes (Figure 1).

As already mentioned, OA is a potent activator of NF-κB in a variety of different cell types [21,22,23,24,25]. We verified this in HepaRG cells. Our findings show an activation of NF-κB in a concentration-dependent manner. Phosphorylation of the NF-κB inhibitor IκB is facilitated by the IκB-kinase (IKK), which is itself regulated by PP2A [40,41,42]. The activation of NF-κB thereby appears to be a direct consequence of the inhibition of PP2A by OA.

NF-κB plays a major role in inflammation. Its target genes include several cytokines, such as IL-1, IL-6, IL-8, and TNFα [26,43]. Our findings show an increase in cytokine release and expression after OA exposure. Inhibited NF-κB activation through NF-κB-inhibitors did not result in an inhibition of translocation of p65 but led to a decreased cytokine release. This might be caused by the fact that the NF-κB-inhibitor JSH-23 is known to inhibit the transcriptional activity of NF-κB and does not only exert its effect by affecting the translocation of the transcription factor [44]. The second NF-κB-inhibitor used, Methysticin, also did not show an inhibition of NF-κB translocation. However, our data also suggest an inhibition of the transcriptional activity of NF-κB, which is in accordance with previous studies [45,46]. However, kinetic aspects might also play a role here. Based on this, the data suggest a NF-κB-dependent activation of cytokine release upon OA exposure in HepaRG cells.

The cytokine IL-6, a major NF-κB target and mediator of inflammatory processes, binds to its respective cytokine receptor to activate JAKs and, subsequently, STATs [36]. JAKs are tyrosine kinases that are associated to a cytokine receptor. After binding of the ligand, the receptors dimerize, which leads to an activation of the pathway. JAKs phosphorylate each other, which enables the binding of STATs. STATs are also phosphorylated and thereby activated. They then dimerize and translocate into the nucleus where they act as transcription factors [36,37] (cp. Figure 1). Our results show an activation of STAT3 in HepaRG cells after exposure to OA, as evidenced in the confocal microscopy experiments. This was confirmed by Western blotting. Simultaneous incubation with JAK inhibitors reversed this activation, which further strengthens the key role of STAT3 in OA-mediated toxicity.

Tanner et al. demonstrated that elevated levels of the cytokine IL-6 in human liver cells lead to a decrease in PXR and constitutive androstane receptor (CAR) activity, two nuclear receptors that play a major role in CYP expression after activation by xenobiotics and dimerization with the RXRα [39]. Furthermore, the activated JAK/STAT signaling pathway can influence RXRα. RXRα plays a central role in the regulation of CYP enzymes, as it associates with many other transcription factors influencing CYP expression [27].

It has been shown that an increased level of proinflammatory cytokines leads to a decreased expression of various drug-metabolizing enzymes and reduced CYP activity [47]. In particular, CYP3A4 is known to be downregulated during inflammation and sepsis. Furthermore, OA is able to strongly downregulate PXR, a main regulator of CYP3A4 [48]. There is clear evidence that PXR regulation occurs through NF-κB and subsequent cytokine signaling [49]. This suggests a connection between NF-κB and CYP3A4 via cytokine release, JAK/STAT and PXR. The CYP1A family (1A1, 1A2, and 1B1), on the contrary, is regulated through the aryl hydrocarbon receptor (AhR). While CAR and PXR can dimerize with RXRα, AhR dimerizes with the aryl hydrocarbon nuclear translocator (ARNT) protein before targeting the regulatory region of their respective target genes [50,51,52]. RNA expression of AhR is upregulated after exposure to OA [20]. AhR expression is controlled by NF-κB and it is able to directly influence the activation of STAT3 [49]. Therefore, a connection between JAK signaling and AhR-mediated CYP expression can be assumed, which remains to be elucidated in future studies.

## 5. Conclusions

Our findings demonstrate a connection of the proposed OA/NF-κB/JAK/STAT pathway to the downregulation of CYP enzymes already reported in an earlier study [20]. OA was able to downregulate the expression of CYP enzymes. With the addition of NF-κB inhibitors or JAK inhibitors, this effect was reversed. This clearly shows a connection between inhibited CYP expression in OA-exposed cells and activation of NF-κB and JAK/STAT signaling by OA. Future research will help to elucidate the consequences of these findings with respect to the inhibition of the metabolic barrier function of the liver and intestine in organisms exposed to OA.

## Figures and Tables

**Figure 1 cells-12-00770-f001:**
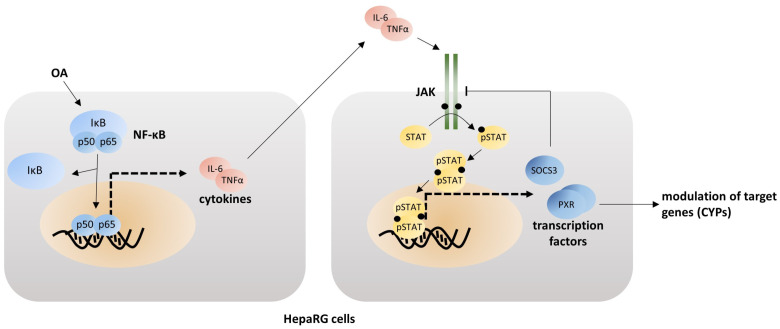
Proposed pathway for the interaction of OA with xenobiotic metabolism in HepaRG cells. OA is able to activate NF-κB, which leads to the translocation of p50 and p60 into the nucleus, where they act as transcription factors, leading to the expression of cytokines. The cytokines, such as IL-6 or TNFα, are released into the surrounding medium by the cells, where they can bind to a cytokine receptor of the same or a neighboring cell. The cytokine receptor is associated with a JAK. Binding of the cytokine leads to dimerization of the receptors, which in turn leads to the JAKs phosphorylating each other. STATs are then able to bind to the JAK, where they are also phosphorylated. Phosphorylated STATs dimerize and translocate into the nucleus, where they can act as a transcription factor to other proteins influencing the CYPs and to SOCS3, a direct target of the JAK/STAT signaling pathway, which is able to deactivate JAK signaling in a feedback loop.

**Figure 2 cells-12-00770-f002:**
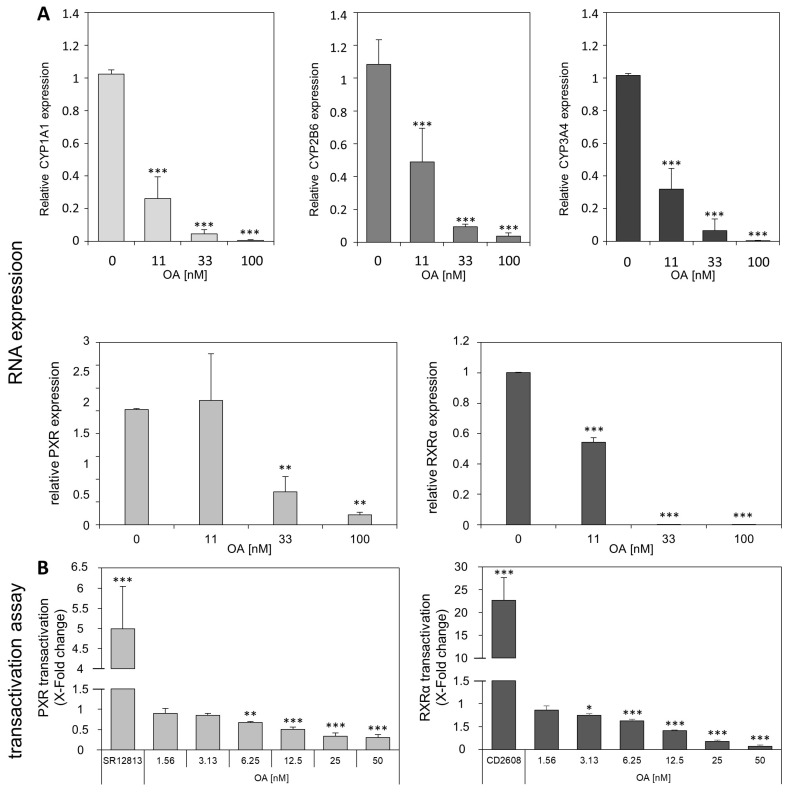
Relative expression of CYP1A1, CYP2B6 and CYP3A4, PXR, and RXRα. (**A**): Differentiated HepaRG cells were treated with 11, 33, and 100 nM OA or with the respective solvent control for 24 h. Analysis of RNA levels of CYP1A1, CYP2B6, CYP3A4, PXR, and RXRα in HepaRG cells after exposure to OA was performed by qPCR. The bar charts show the resulting fold changes as mean of three independent replicates, relative to the solvent control. (**B**): Transactivation of PXR and RXRα in HEK-T cells. The cells were transfected with plasmids before incubation with OA for 24 h. SR12813 (10 µM) and CD2608 (100 nM) were used as positive controls. The cytotoxicity assay in HEK-T cells relevant for the chosen OA concentrations for the transactivation assay can be found in Appendix A. The bar charts show the resulting fold changes as mean of three independent replicates, relative to the solvent control. Statistical analysis for all charts (n = 3) was performed using one-way ANOVA followed by Dunnett’s post-hoc test (* *p* < 0.05; ** *p* < 0.01; *** *p* < 0.001).

**Figure 3 cells-12-00770-f003:**
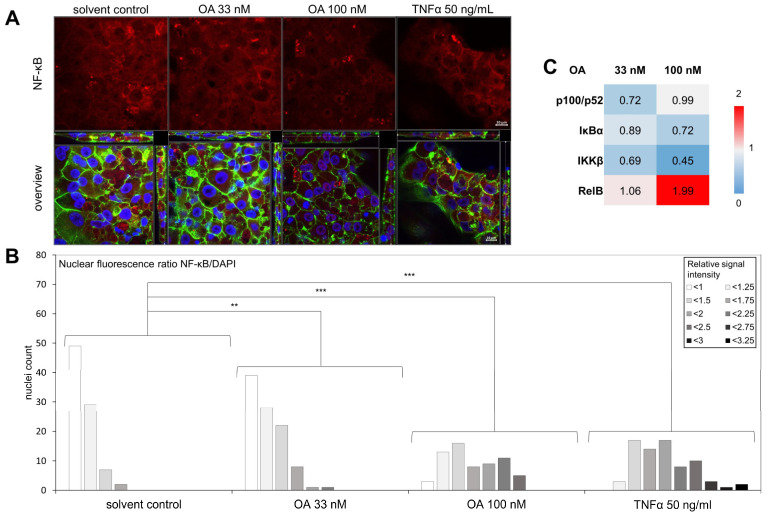
Activation of NF-κB in HepaRG cells. Differentiated HepaRG cells were treated with 33 and 100 nM OA alone or 33 µM OA in combination with 30 µM JSH-23, or 30 µM Methysticin, two NF-κB activation inhibitors, or with the respective solvent control for 24 h. (**A**): Nuclei were stained with DAPI, and the actin cytoskeleton was stained using ActinGreen™ 488 ReadyProbes™ reagent. Immunostaining of NF-κB was carried out using a primary antibody against NF-κB subunit p65 and stained using a secondary antibody conjugated with the fluorophore Alexa Fluor™ 633. Fluorescence was detected using a confocal laser scanning microscope at ex wavelengths of 405 nm (DAPI, blue), 488 nm (ActinGreen, green), and 633 nm (NF-κB, red). Z-stacks spanning through the entire cell layer were recorded at 63× magnification. Brightness was increased by 15% for the green channel, 60% for the blue channel, and 70% for the red channel in each image. (**B**): The nuclear fluorescence ratio was determined using ImageJ 1.53e. Nuclei were marked as ROI and the fluorescence intensity of the red and blue channels in each ROI was determined separately. A ratio of NF-κB/DAPI signal intensity was then calculated and normalized to the mean of the solvent controls. The nuclei were divided into groups based on their signal intensity. Results show the number of nuclei per signal intensity group for each treatment group. Between 65 and 99 nuclei were evaluated per condition. Statistical analysis was performed using Wilcoxon rank-sum test (** *p* < 0.01; *** *p* < 0.001). (**C**): Results of the DigiWest analysis for p100/p52, IκBα, IKKβ, and RelB. Results were obtained using 3 pooled replicates.

**Figure 4 cells-12-00770-f004:**
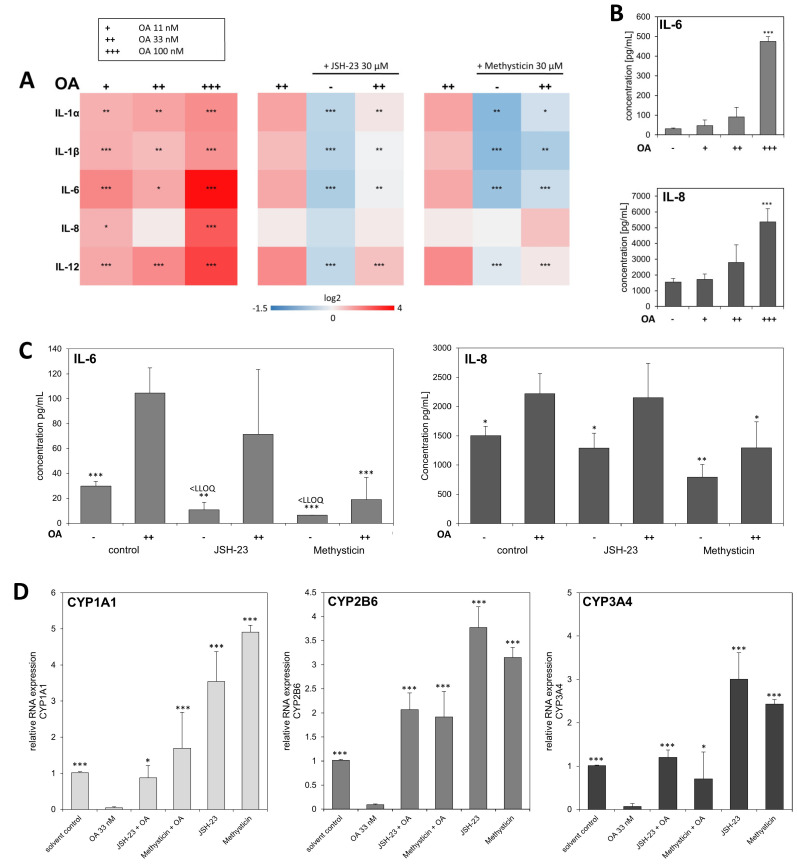
RNA expression and release of Interleukins in HepaRG cells after exposure to OA. Differentiated HepaRG cells were treated with 11, 33, and 100 nM OA alone or 33 µM OA in combination with 30 µM JSH-23, or 30 µM Methysticin, two NF-κB activation inhibitors, or with the respective solvent control for 24 h. (**A**): Analysis of mRNA levels of interleukins was performed by qPCR. The heatmap shows the log2 values of the resulting fold changes as mean of six (heatmap left) or three independent replicates, relative to the solvent control. Statistical analysis (n = 6, n = 3) was performed against the solvent control (heatmap left) or the 33 nM OA sample (rest) using one-way ANOVA followed by Dunnett’s post-hoc test (* *p* < 0.05; ** *p* < 0.01; *** *p* < 0.001). (**B**): Release of IL-6 and IL-8 from HepaRG cells into the cell culture supernatant after exposure to OA. The IL-content in the cell culture supernatant was quantified using Luminex multiplex assay. Statistical analysis (n = 3) was performed against the solvent control using one-way ANOVA followed by Dunnett’s post-hoc test (* *p* < 0.05; ** *p* < 0.01; *** *p* < 0.001). LLOQ IL-6: 12.89 pg/mL; LLOQ IL-8: 2.54 pg/mL. (**C**): Release of IL-6 and IL-8 from HepaRG cells into the cell culture supernatant after exposure to OA in combination with the inhibitors. The content in the supernatant was analyzed as described in Figure 2B. (**D**): Analysis of RNA levels of CYP1A1, CYP2B6, and CYP3A4 by qPCR in HepaRG cells after exposure to OA and the inhibitors. The bar charts show the resulting fold changes as mean of three independent replicates, relative to the solvent control. Statistical analysis for both (**C**) and (**D**) (n = 3) was performed against the 33 nM OA sample using one-way ANOVA followed by Dunnett’s post-hoc test (* *p* < 0.05; ** *p* < 0.01; *** *p* < 0.001).

**Figure 5 cells-12-00770-f005:**
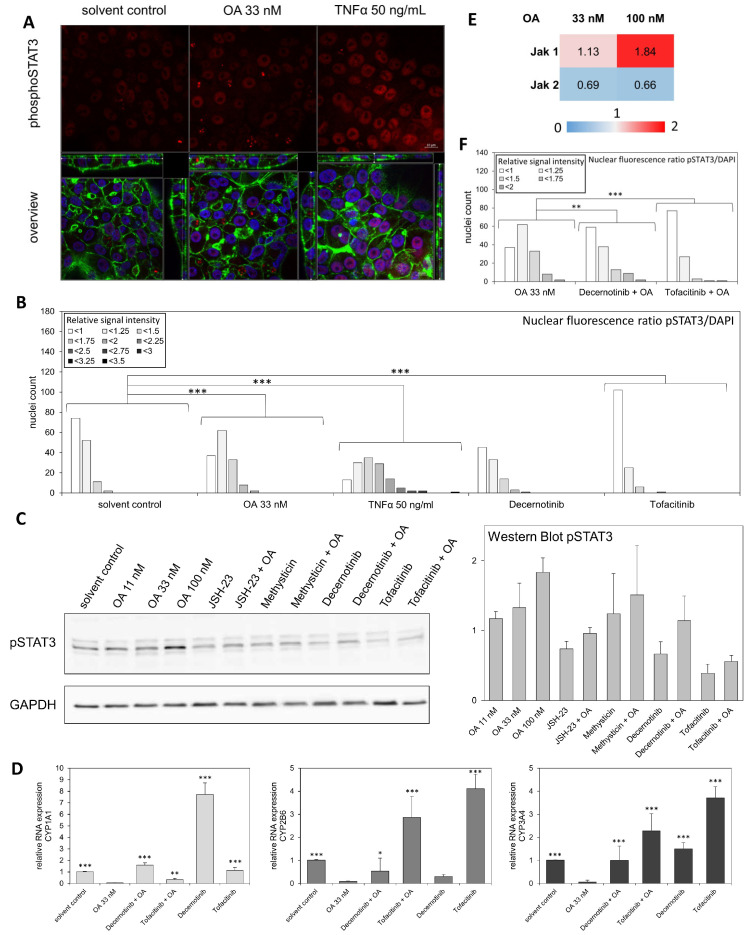
Activation of JAK/STAT signaling in HepaRG cells after exposure to OA. Differentiated HepaRG cells were treated with 33 and 100 nM OA alone or 33 µM OA in combination with 40 µM Decernotinib, or 40 µM Tofacitinib, two JAK activation inhibitors or with the respective solvent control for 24 h. (**A**): Nuclei were stained with DAPI, and the actin cytoskeleton was stained using ActinGreen™ 488 ReadyProbes™ reagent. Immunostaining of STAT3 was carried out using a primary antibody against phosphorylated STAT3 and stained using a secondary antibody conjugated with the fluorophore Alexa Fluor™ 633. Fluorescence was detected using a confocal laser scanning microscope at ex wavelengths of 405 nm (DAPI, blue), 488 nm (ActinGreen, green), and 633 nm (phosphoSTAT3, red). Z-stacks spanning through the entire cell layer were recorded at 63× magnification. Brightness was increased by 40% for the blue and red channels in each image. (**B**): The nuclear fluorescence ratio was determined using ImageJ 1.53e. Nuclei were marked as ROI and the fluorescence intensity of the red and blue channels in each ROI was determined separately. A ratio of pSTAT3/DAPI signal intensity was then calculated and normalized to the mean of the solvent controls. The nuclei were divided into groups based on their signal intensity. Results show the number of nuclei per signal intensity group for each treatment group. Between 96 and 142 nuclei were evaluated per condition. Statistical analysis was performed against the solvent control using Wilcoxson rank-sum test (*** *p* < 0.001). (**C**): Western blot of lysed HepaRG cells against phosphorylated STAT3. Cells were incubated as described above with the addition of 30 µM JSH-23 and 30 µM Methysticin alone or in combination with 33 nM OA. Western blot were obtained using the wet blot method. Three biological replicates were independently analyzed and normalized against the housekeeper GAPDH. Afterwards, all samples were normalized against their respective solvent control. (**D**): Analysis of RNA levels of CYP1A1, CYP2B6, and CYP3A4 by qPCR in HepaRG cells after exposure to OA and the inhibitors. The bar charts show the resulting fold changes as mean of three independent replicates, relative to the solvent control. Statistical analysis (n = 3) was performed against the 33 nM OA sample using one-way ANOVA followed by Dunnett’s post-hoc test (* *p* < 0.05; ** *p* < 0.01; *** *p* < 0.001). (**E**): DigiWest results for JAK1 and JAK2. Three independent replicates were incubated and combined before measuring. (**F**): The nuclear fluorescence ratio was determined as in (**B**). Between 109 and 142 nuclei were evaluated for each condition. Statistical analysis was performed against the 33 nM OA sample using Wilcoxson rank-sum test (** *p* < 0.01; *** *p* < 0.001).

## Data Availability

Data is contained within the article or Appendix A. Additional data upon request.

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
