# Peer review of "Okadaic Acid Activates JAK/STAT Signaling to Affect Xenobiotic Metabolism in HepaRG Cells"

_cells, 2023, doi:10.3390/cells12050770_

Round 1
Reviewer 1 Report
The authors have investigated the molecular mechanism of the effect of okadaic acid (OA), a marine biotoxin, on the expression of the cytochrome P450 enzymes CYP1A1, CYP2B6 and CYP3A4, and the expression and transactivation of transcription factors using HepaRG or HEK-T cells, respectively.
Based on the results obtained before and those achieved in the present work, they have proposed that the downregulation of CYP2B6 and CYP3A4 by OA proceeds via the NF- κB → cytokines → JAK/Stat → transcription factors (PXR, RXRα) pathway, while the molecular mechanism of CYP1A1 regulation (engaging the AhR transcription factor) by OA requires additional studies.
Minor points:
1. The authors should indicate in the Abstract which results were obtained in the present work and emphasize the novelty of their present achievements.
2. In the Discussion, differences in the physiological molecular mechanisms of CYP1A1 and CYP2B6/CYP3A4 regulation could be described in greater detail.
The authors have investigated the molecular mechanism of the effect of okadaic acid (OA), a marine biotoxin, on the expression of the cytochrome P450 enzymes CYP1A1, CYP2B6 and CYP3A4, and the expression and transactivation of transcription factors using HepaRG or HEK-T cells, respectively.
Based on the results obtained before and those achieved in the present work, they have proposed that the downregulation of CYP2B6 and CYP3A4 by OA proceeds via the NF- κB → cytokines → JAK/Stat → transcription factors (PXR, RXRα) pathway, while the molecular mechanism of CYP1A1 regulation (engaging the AhR transcription factor) by OA requires additional studies.
Minor points:
1. The authors should indicate in the Abstract which results were obtained in the present work and emphasize the novelty of their present achievements.
2. In the Discussion, differences in the physiological molecular mechanisms of CYP1A1 and CYP2B6/CYP3A4 regulation could be described in greater detail.
Author Response
- The authors should indicate in the Abstract which results were obtained in the present work and emphasize the novelty of their present achievements.
A sentence was added to the abstract emphasizing the novelty of the present work.
- In the Discussion, differences in the physiological molecular mechanisms of CYP1A1 and CYP2B6/CYP3A4 regulation could be described in greater detail.
We added a section in the discussion, mentioning the binding of CAR and PXR to RXRα and on the contrary the binding of Ahr to ARNT.
Reviewer 2 Report
In this study, the authors investigated a possible underlying mechanism of the downregulation of cytochrome P450 (CYP) enzymes and the nuclear receptors PXR and RXRα by Okadaic acid (OA) which is a marine biotoxin, through NF-κB and subsequent JAK/STAT activation in human HepaRG hepatocarcinoma cells.
1. The introduction appeared as a discussion. It needs to be re-written focusing on the research problem.
2. The objective should be written in a clear way.
3. Few lines in the introduction are repeated in the discussion.
4. A proper conclusion should be added in the manuscript.
5. Expression of cytochromes (CYPs) should be determined through western blot.
6. Novelty of the work should be described properly.
7. The manuscript must be written in a clear concise way and all the grammatical errors needs to be corrected.
Author Response
- The introduction appeared as a discussion. It needs to be re-written focusing on the research problem.
The introduction was shortened and is now focused on the main points. Furthermore, sections more suited for the discussion were moved from the introduction to the discussion.
- The objective should be written in a clear way.
The objective was rewritten, we hope it is more clear now.
- Few lines in the introduction are repeated in the discussion.
We removed or rewritten all sentences that were repeated in the Discussion.
- A proper conclusion should be added in the manuscript.
A conclusion was added to the manuscript.
- Expression of cytochromes (CYPs) should be determined through western blot.
We thank the reviewer for his/her suggestion. We have already published the RNA and protein expression of the CYP enzymes in our previous work (https://doi.org/10.17179/excli2022-5033, page 1058): “Data at the mRNA level were complemented by mass-spectrometric determination of selected CYPs at the protein level following an identical experimental approach for treatment with OA and CYP inducers. Comparable to what had been observed at the mRNA level, exposure of HepaRG cells to OA led to diminished levels of most CYPs”
Additionally, we also examined the CYP activity after exposure to OA in this previous publication (page 1058): “Comparable to the observations at the mRNA and protein levels, CYP activities were reduced by OA treatment”
A sentence making our previous results clear was added to the discussion.
- Novelty of the work should be described properly.
We added a section in the introduction and in the discussion, emphasizing the novelty of our work and how it can be distinguished from other studies.
- The manuscript must be written in a clear concise way and all the grammatical errors needs to be corrected.
We have carefully revised grammar and orthography.